# Extended Depth of Focus Two-Photon Light-Sheet Microscopy for In Vivo Fluorescence Imaging of Large Multicellular Organisms at Cellular Resolution

**DOI:** 10.3390/ijms241210186

**Published:** 2023-06-15

**Authors:** Takashi Saitou, Takeshi Imamura

**Affiliations:** 1Department of Molecular Medicine for Pathogenesis, Graduate School of Medicine, Ehime University, Toon 791-0295, Ehime, Japan; timamura-ind@umin.ac.jp; 2Translational Research Center, Ehime University Hospital, Toon 791-0295, Ehime, Japan

**Keywords:** light-sheet microscopy, extended depth of focus, two-photon excitation, in vivo imaging

## Abstract

Two-photon excitation in light-sheet microscopy advances applications to live imaging of multicellular organisms. In a previous study, we developed a two-photon Bessel beam light-sheet microscope with a nearly 1-mm field of view and less than 4-μm axial resolution, using a low magnification (10×), middle numerical aperture (NA 0.5) detection objective. In this study, we aimed to construct a light-sheet microscope with higher resolution imaging while maintaining the large field of view, using low magnification (16×) with a high NA 0.8 objective. To address potential illumination and detection mismatch, we investigated the use of a depth of focus (DOF) extension method. Specifically, we used a stair-step device composed of five-layer annular zones that extended DOF two-fold, enough to cover the light-sheet thickness. Resolution measurements using fluorescent beads showed that the reduction in resolutions was small. We then applied this system to in vivo imaging of medaka fish and found that image quality degradation at the distal site of the beam injection could be compensated. This demonstrates that the extended DOF system combined with wide-field two-photon light-sheet microscopy offers a simple and easy setup for live imaging application of large multicellular organism specimens with sub-cellular resolution.

## 1. Introduction

Light-sheet microscopy is fluorescence microscopy in which the detection and illumination optical axes are placed orthogonally. It achieves high-resolution biological imaging with a thin sheet of light from the side on the focal plane of the detection objective lens [1]. This technology is advantageous in achieving high spatial resolution in the direction of the optical axis (*z*) and offers a higher acquisition speed with lower phototoxicity because, due to this thin selective plane illumination, the total fluorophore excitation is greatly reduced compared to conventional wide-field microscopy. Therefore, this technology has become a powerful tool for time-lapse observations over long periods of time, enabling the elucidation of single-cell behaviors of multicellular organism morphogenesis [2,3].

The illumination and the detection machinery determine the optical properties of the light-sheet microscopy, such as spatial resolution and field of view (FOV). There is a trade-off between FOV and axial resolution due to the orthogonal geometry of the illumination and detection pathways of light-sheet microscopy. Since the excitation light is incident from the side, thus, the length of the focused beam is directly related to the FOV, and the width of the beam is directly related to the axial resolution. This means that lengthening the beam to improve FOV results in a thicker beam and lower resolution, whereas narrowing the beam to improve resolution results in a shorter beam and narrower FOV. This makes high-resolution imaging of large specimens challenging.

The use of non-diffracting beams, such as Bessel [4,5,6,7], Airy [8], and lattice [9] beams, has been proposed to address the trade-off between FOV and axial resolution in light-sheet microscopy. A Bessel beam is formed by injecting a ring-shaped beam into a lens, resulting in the central beam shape becoming elongated along the propagation axis like a needle. This elongated shape extends the focus depth of excitation, making it effective in improving both FOV and resolution. In addition, the self-healing property of the Bessel beam improves the shadows behind the object caused by the shielding and scattering of the laser beam. However, there are trade-offs when using Bessel beams. The distribution of energy to the side lobes is related to the beam length in that suppressed energy in the side lobes generates relatively short propagation beams, and vice versa [7,10], and the generation of out-of-focus fluorescence signals due to side lobes reduces the signal-to-noise ratio. One way to address this issue is to use two-photon excitation, because the quadratic dependence of the fluorescence signal of two-photon excitation reduces the contribution to the signal from the side lobes relative to the central lobe. Furthermore, two-photon excitation technology is suitable for the imaging of living animals because it can be caused by near-infrared light with a wavelength of 700–1300 nm, which is less toxic to living organisms. Near-infrared light is also more permeable to living organisms than visible light, making it suitable for observing deep inside the organism [11,12]. Two-photon excitation in light-sheet microscopy has the advantages of improved penetration depth and background rejection [13]. The use of a two-photon Bessel beam extends the propagation length while maintaining a thin beam width. Several applications of two-photon–excited Bessel beams have been reported in the subcellular structures of living organisms [10], multicellular organisms of C. Elegans [6], zebrafish [14], and tumors [4,7]. We previously achieved Bessel beams of full width at half maximum (FWHM) of 600–1000 μm with less than a 5-μm thickness when using a 10× magnification NA 0.3 illumination objective lens, and developed a light-sheet microscope that made it possible to observe a whole live medaka fish (~1 mm) for a long period (~3 days) at cellular resolution [15].

The design of the light-sheet microscope should match the thickness of the illumination sheet and the depth of focus (DOF) of the detection objective (DO). High numerical aperture (NA) objectives can efficiently collect fluorescence signals, but they have a tight DOF. This mismatch between the light-sheet and the DOF of the DO can cause not only a loss of in-focus fluorescence signals, but also the detection of out-of-focus signals as backgrounds, reducing the signal-to-noise ratio (SNR). Moreover, when the illumination light propagates through biological samples, due to the inhomogeneous distribution of objects, bending and spreading of the light-sheet by scattering and absorption through samples leads to the lowering of signals and degrading of image contrast, decreasing the penetration deep inside tissues. For two-photon imaging over large samples, this light-sheet modulation can become a severe issue because of its quadratic dependence on two-photon excitation. Therefore, achieving high-resolution in vivo imaging over the entire sample is challenging.

An extension of DOF can address this issue of illumination and detection mismatch. In this study, we used this technique to improve the wide-field two-photon microscopy for higher resolution and SNR imaging while keeping a large FOV. For this purpose, we used a low magnification (16×) and high NA (0.8) DO that provides a DOF of ~2 μm. This is narrower than the light-sheet thickness of our system, while we previously used a low magnification (10×) and middle NA (0.5) DO that gives a DOF of ~5.4 μm, covering the full light-sheet thickness. To solve the mismatch, we used a stair-step device composed of five-layer annular zones. The device has been reported to extend DOF by several times more than the conventional system [16]. The DOF extension by the stair-step device is limited up to several folds, but a small mismatch between the detection objective and the light-sheet, such as in our case, was successfully eliminated. We showed that the effective collection of the fluorescence signal can be achieved, and the resolution measurement using fluorescent beads clearly showed that the high resolutions (lateral ~1 μm and axial ~2–3 μm) are maintained over the FOV. The reduction of the lateral resolution is small, and an almost equivalent axial resolution is achieved by this device. We further applied this system to in vivo imaging of medaka fish. We observed that image quality degradation at the distal side of the beam injection could be compensated, demonstrating that the microscope with the extended DOF device allows imaging of living fish (medaka) embryos with a sub-cellular resolution over the entire body. The extended DOF system combined with wide-field two-photon light-sheet microscopy offers a simple and easy setup for the live imaging application of large-size multicellular organism specimens in sub-cellular resolution.

## 2. Results

### 2.1. Construction of Two-Photon Light-Sheet Microscopy with a DOF Device

We constructed a two-photon light-sheet microscope with an extended DOF device that stretches the point spread function (PSF) of the detection system along the axial (*z*) direction (Figure 1A). The system is based on the two-photon excitation digital scanned light-sheet microscope that we previously reported [15]. The illumination optics are based on a Bessel beam created through a combination of axicon and convex lenses, generating longer extended thin beams (600–1000-μm length, 3–4-μm thickness). The infrared Gaussian beam with a wavelength of 920 nm is transformed through the Bessel beam-forming unit, and the light-sheet is generated by scanning the beam along the *y*-axis. The illuminated plane is imaged through the DO, tube lens, and an additional 4f imaging unit. In order to achieve a wide FOV and high-resolution imaging at the same time, lower magnification with a high NADO, CFI75 LWD 16× W (16× magnification, NA = 0.8) was used. A theoretical criterion of the DOF for this DO was calculated as d = λ/n(1 – cos α) =1.96 μm, where λ (525 nm) is the wavelength, n (1.33) is the refractive index, and α is the angle determined by the objective NA (0.8) [17]. The beam thickness was slightly broader than the DOF of DO. This mismatch can be compensated by extending the DOF. Extension of the DOF was achieved through the insertion of a circularly symmetric stair-step device into the second pupil plane of the microscope system (Figure 1A). The stair-step device is composed of five-layer annular zones, and it has been reported to extend the DOF several times with a small reduction of lateral resolution [16] (Figure 1B). 

This layered phase mask divides the pupil into five sub-apertures and creates an axially elongated PSF. Using a computer simulation of the scalar Debye theory [18], we calculated PSFs for both the microscope system with (extended DOF) and without (control) the devices (Appendix B). The simulated PSFs demonstrated that the DOFs (FWHM along the *z*-axis) of the systems with and without the device are 3.55 μm and 1.73 μm, respectively, and it indicates that the device extends the DOF two-fold (Figure 2).

### 2.2. Performance of Extended DOF Two-Photon Light-Sheet Microscopy

To compare the fluorescence signal collection ability of two-photon excitation volumes of the Bessel beams, we measured beam line profiles using a fluorescein solution (10 μg/mL) (Figure 3A). The peak intensity of the beams detected with the device is about half of that detected without the device. To quantitatively compare these profiles, we extracted the line profiles along the longitudinal direction (*x*) and transverse (*y*) direction at the beam waist (Figure 3B,C). The estimated beam lengths (FWHM) were almost equal to ~720 μm between the control and extended DOF systems, but signal collection efficiency differed at the distal site of the beam propagation—relative intensity for the extended DOF is higher than that for the control at that region (Figure 3B). This comes from the relation between the beam thickness and DOF. Hence, to evaluate the beam thickness, we next investigated the transverse line profiles at the beam focus, which showed that the thickness with the device is slightly broader than that without the device (Figure 3C). Since these thicknesses are from the observed 2D images, the quantified values (FWHM that is denoted as FWHM2D) do not coincide with actual 3D beam thicknesses, which should be considered using a 3D analysis. Thus, in order to perform a more accurate evaluation of the beam thickness, we developed a way to infer the actual 3D FWHMs using simulations (Appendix A). We first set Gaussian beam lines with an arbitrarily defined FWHM thickness as a 3D object. Then, 3D images of the beam lines were obtained by convolutions of the Debye PSFs (with and without the device) and the beam profiles. Finally, the observed 2D images were created by the sum of the 3D image along the *z*-axis. The result showed linear relationships between these actual FWHMs and FWHM2D (Appendix A). Using these, we calculated the FWHMs from images (Figure 3D and Appendix A). The beam thicknesses were 3–4.5 μm, depending on the location along the longitudinal axis (center ~3 μm, proximal ~3–3.5 μm, distal ~3.5–4.5 μm). These thicknesses were confined within the extended DOF, but not confined in the control DOF. The thickness for the extended DOF was slightly larger than that for the control, and its ratios were 1–1.1 (Figure 3E). This is in good agreement with the reported lateral resolution reduction [16], indicating that the difference in FWHM comes from resolution degradation by the device. Therefore, the DOF of the system for the control system is narrower than the beam thickness, indicating that losses of fluorescence signal collection exist, while that of the extended DOF system matches the beam thickness, indicating effective fluorescence signal collection, which is particularly recognizable at the distal site signal intensities (Figure 3B). In addition, calculations of the coefficient of variation of longitudinal line intensities over the full FOV resulted in 0.43 (control) and 0.40 (exDOF), which indicated an increased homogeneity of the fluorescence signal collection in the extended DOF system. These results suggest that decreased out-of-focus fluorescence signal in the extended DOF system increases the image homogeneity and makes the fluorescence signal collection more effective. In order to investigate the resolution of the microscope, we measured PSFs based on fluorescent beads of 200 nm diameter (Figure 4 and Appendix A). In order to investigate the positional variability of the resolution, we created cropped images of 512 × 1536 pixels at the areas of the center, proximal side, and distal side of beam injection in the images (Figure 4A,C,E). We then evaluated the beads’ FWHMs for both the lateral and axial directions (Figure 4B,D,F). The extended DOF system showed worse lateral resolution than the control system for all three different areas, which is consistent with the beam line profile results. The axial FWHMs for the extended DOF remained nearly unchanged compared with the control at the center and proximal sides, while those for the extended DOF at the distal side were slightly larger than the control. The reason for these less significant changes appears to be related to the beam thickness and geometry of light-sheet microscopy. The resolution of light-sheet microscopy is determined by both the illumination and detection systems. The beam thicknesses at the center and proximal areas are 3–3.5 µm, which are confined within the extended DOF, and are larger than the control DOF, but not significantly larger. In these cases, the beam energy is relatively concentrated within the in-focus area. Thus, the effect of the DOF extension is not substantial. On the other hand, when the beam thickness at the distal side increases to >3.5 μm, the axial resolution may become highly dependent on the detection DOF. Consequently, the difference in axial FWHMs becomes more pronounced at the distal side. Therefore, since the resolution reduction can be kept to a minimum, it is confirmed that high resolution in the axial and lateral directions is largely maintained throughout the FOV.

### 2.3. In Vivo Imaging Application of the Extended DOF Two-Photon Light-Sheet Microscopy

We have thus far demonstrated that homogeneous imaging over the full FOV is possible for the extended DOF system. To test the performance of the extension of DOF for in vivo imaging, we imaged five day-post-fertilization (dpf) embryos of the medaka FLT4-EGFP strain, which expresses EGFP in the lymphatic endothelial cells and some venous vascular endothelial cells [19]. Embryos were raised at 28 °C artificial seawater with methylene blue to the developmental stage. The samples were dechorionated using the hatching enzyme prior to the microscope observation, and were embedded in 1% agarose gel in the embryonic culture solution for microscopy imaging. We acquired the *z*-stack images with 2-μm step size, 1024 × 1024 pixels (0.8 μm × 0.8 μm/pixel), and 100-ms exposure time. The *z*-stack images consist of optical sections spanning > 600 μm in depth. The body trunk of the lymphatic vessel (proximal side in the images in Figure 5A, blue circle) and veins surrounding the yolk (distal side in the images in Figure 5A, green circle) can be clearly recognized in the maximum intensity projection of the 3D stack data. This indicated that high SNR imaging was possible in both the control and the extended DOF system. By comparing the images of the control and extended DOF, resolution and structures in the proximal (right side in the images) and center area have no clear differences. However, there are differences in the distal (left) side of the images, as indicated by the orange rectangle (Figure 5A). Sharper and brighter venous shapes were observed there in the extended DOF case, while blurred shapes were observed in the control case. This result showed that the extended DOF system can compensate for image quality degradation. To investigate whether effective fluorescence correction exists in in vivo imaging at the distal side, similar to the fluorescence solution experiment, we evaluated an intensity profile from the live sample images. We extracted the line profiles for several selected slices and calculated the ratio of the proximal to distal intensity (Figure 5B). The averages of the ratios with the extended DOF were larger than that of the control. The sign test of these ratio data showed statistically significant differences. These results indicated that image quality improvement at the distal site of the field of view by extending DOF can be recognized in in vivo imaging.

## 3. Discussion

This study investigated the utility of the five-layered stair-step device as an extension of the DOF to achieve a wide FOV and high-resolution imaging in two-photon excitation Bessel beam light-sheet microscopy. The results showed that the lateral resolution reduction is kept small, and image quality degradation at the distal side of the beam injection can be compensated, demonstrating that the microscope with the extended DOF device allows imaging of living fish (medaka) embryos with high resolution over the entire body. The extended DOF system combined with wide-field two-photon light-sheet microscopy offers a simple and easy setup for live imaging of large-size multicellular organism specimens in sub-cellular resolution.

### 3.1. Design of Light-Sheet Microscopy

Light-sheet microscopy employs separate illumination and detection pathways in which the pathways are placed orthogonally to each other. Due to this optical geometry, the spatial resolution is determined by a combination of illumination and detection components, which should thus be carefully designed. The illumination part of our microscope system is based on the two-photon excitation Bessel beam, which achieves 600–1000-μm FOV with 3–5-μm thickness when using a NA 0.3 10× magnification illumination objective lens [15]. A Bessel beam is non-diffractive, and its shape is needle-like along the propagation axis, greatly extending the excitation length while maintaining its thin beam waist. Two-photon excitation also suppresses the side lobes of the Bessel beams, enabling high-contrast imaging. For the detection part, we selected a high NA, low magnification objective lens (NA 0.8, 16× magnification) to simultaneously achieve wide-field and high-resolution imaging. The excitation beam covers nearly the full range of the FOV (832 μm), and a high NA objective lens efficiently collected illuminated fluorophore signals. However, this high NA objective lens has a narrow DOF of ~1.73 μm estimated by simulation. This is narrower than the beam thickness. The use of the stair-step device extended the DOF by two-fold to 3.55 μm, which matched the illumination beam, and thus the effective collection of the fluorescence signal can be achieved in this high-resolution setup. The resolution measurement using fluorescent beads clearly showed that the high resolutions (lateral ~1 μm and axial ~2–3 μm) are kept over the FOV. The spatial resolution of light-sheet microscopy is determined by a combination of illumination and detection systems. Therefore, both the detection objective and the illumination beam thickness contribute to the PSFs. The measured axial PSFs match this beam thickness as expected. The system with the extended DOF provides slightly lower quality lateral resolution and almost equivalent axial resolution compared with the system without the device. This result suggests that the DOF system extended by the stair-step device provides a convenient and easy setup for wide-field and high-resolution two-photon light-sheet microscopy.

An application of the stair-step device in light-sheet microscopy has been reported [20]. They applied this extended DOF method to the detection of the high NA (0.8) and high magnification (40×) DO to match the thicker light-sheet (~8 μm) volume, and improved a way to axially elongate the large volume detection, although information on the axial position was compromised. Our strategy to utilize the extended DOF technique differs from that report. We used the device to solve a mismatch between a two-photon excitation light-sheet and the DOF of DO, and achieved high-quality imaging while keeping a large FOV without compromising the axial resolution.

There are other methods for extending the DOF, such as the use of wavefront coding techniques [21,22] and an axicon lens [23]. The wavefront coding technique requires computational image processing that performs the reconstruction of volumetric images from detected plane images. This is advantageous to 3D imaging, but it costs computational demand, and additionally spatial resolution is limited. The use of axicon in the detection pathway is an alternative approach that does not need computational image reconstruction. This elongates the DOF by 20-fold with a small reduction in the lateral resolution, which has been applied to sheet-scanning light-sheet microscopy for fast volumetric imaging [23]. However, an intensity variation dependence in the axial direction appears. The DOF extension is limited in the stair-step device, but both a small mismatch between the illumination and detection machinery of the light-sheet system and the misalignment of the light-sheet through living samples are well suited for this device.

### 3.2. Application to In Vivo Imaging

As discussed above, the illumination Bessel beam offers highly homogeneous illumination over the FOV of the microscope system while keeping its thickness. Combined with a high NA objective, it further provides high-resolution, high-contrast imaging for large samples. When the illumination light propagates through the biological samples, bending and spreading of the light-sheet by scattering and absorption through samples leads to the lowering of signals and degrading of image contrast, decreasing the penetration deep inside tissues. For two-photon imaging over large samples, this light-sheet modulation is a severe issue because of its quadratic dependence on two-photon excitation. It is expected that the original system (without the device) is sensitive to these light-sheet modulations. Actually, our results showed the image quality degradation for living samples at the distal site from the beam injection. We demonstrated image quality recovery at the distal site in the extended DOF system, although it provides similar image quality to that of the conventional setup (without the device) at the center and proximal areas of the images. Therefore, an extension of the DOF is useful for compensation of the illumination beam modulation. Therefore, two-photon light-sheet microscopy with extended depth of focus provides an easy setup for high-quality imaging of living large-sized samples.

### 3.3. Future Application to High Throughput Analysis

Two-photon excitation microscopy is becoming a promising tool for observing living thick and opaque tissues, and has been extensively used as a histopathological diagnostic tool for assessment of various diseases [24]. We have reported the use of multiphoton microscopy for the diagnosis of osteoarthritis, liver fibrosis, and breast tumors [25,26,27] with label-free imaging techniques. In order to extend these digital pathology analyses and implement these techniques clinically, a framework that combines hardware and software to measure and analyze a large number of samples on a daily basis with high throughput is necessary. The light-sheet microscope is suitable for this large-scale analysis due to its rapid imaging speed. Therefore, it is important to develop a 3D pathological analysis system based on a light-sheet microscope capable of 3D measurement at high speed and high resolution.

## 4. Materials and Methods

### 4.1. Optical System

The base optical system has been described previously [15]. For illumination, a femtosecond pulsed infrared laser, InSight DeepSee (Spectra-Physics, MKS Instruments, Inc., Andover, MA, USA) was used. The Insight DeepSee emits pulses with a duration of 120 fs, a repetition rate of 80 MHz over the near-infrared wavelength range of 700–1300 nm, and works with the average power of 1.2 W at a 920-nm wavelength. The power of the incident beam was modulated through HWP and GLP. The beam was transmitted through a Bessel beam-forming unit in which the Gaussian incident beam was expanded to a 1/e2 diameter of 5.74 mm using a beam expander and transmitted to the lens-axicon triplet to create an annulus ring pattern. The tuned beam was transmitted to the illumination objective through the Galvano mirror scanner (Model 6200H, Cambridge Technology, Peachtree Corners, GA, USA), a telecentric fΘ lens (EFL 115 mm) (Edmund Optics, Barrington, NJ, USA, 64422), and a single convex lens (f = 150 mm). The objective lenses used were a 10× magnification dry lens (LMPLN10XIR, 10× magnification, NA = 0.3, Olympus Evident, Tokyo, Japan) for illumination and a 16× magnification water immersion lens (CFI75 LWD 16× W, 16× magnification, NA = 0.8, Nikon, Tokyo, Japan) for detection. A laser (InSight Deepsee) with a 920 nm wavelength was used in all experiments. The average laser power at the back focal plane of the objective is adjusted at 500 mW. The light-sheet was created in the *xy*-plane by scanning the beam along the *y*-direction using a Galvano mirror scanner at a rate of 100 Hz along the *y*-axis. This laser-induced light-sheet was detected through the detection objective lens, a tube lens with a 200-mm focal length (TTL200-A, Thorlabs, Newton, NJ, USA), and an image-splitting optics system, W-View Gemini-2C (A12801-10, Hamamatsu Photonics, Hamamatsu, Japan). The extended DOF device (A12802-35-100, Hamamatsu Photonics) was inserted in the second pupil plane of the optical system. An sCMOS camera with a 6.45-μm pixel size with 2048 × 2048 pixels (Orca flash 4.0 v3, Hamamatsu Photonics) was used for image acquisition. The images were recorded as unsigned 16-bit gray-level images. During the emission pathway, a short-pass filter at 700 nm (Semrock, Saitama, Japan) was inserted to cut the scattered excitation lights, and band-pass filters at 520/35 nm (center wavelength/bandwidth) (Semrock) were used. The sample chamber was made from acrylic plates and cover glasses of 0.12–0.17 mm thicknesses (Matsunami, Osaka, Japan). The sample-mounting holder was placed on the motorized stages (M-111.1DG and M-116.DG, Physik Instrumente, Karlsruhe, Germany), controlling the translation in the *xyz*-directions and the angle rotation in θ along the *y*-axis. The *z*-stack image sequence was captured by moving the motorized stages along the *z*-direction. A program to control the camera, motorized stages, motorized filter wheel, and Galvano mirror scanner was written using Labview2015 software (National Instruments, Austin, TX, USA).

### 4.2. Measurement of Optical Properties

The sample chamber was filled with 10 μg/mL fluorescein solution to measure beam line profiles. In order to evaluate the beam extent and thickness, FWHMs along the longitudinal (*x*) and transverse (*y*) axis were calculated. For evaluation of the microscopy resolution, yellow–green fluorescence beads (FluoSpheres, Thermo Fisher Scientific, Waltham, MA, USA) of a 200-nm diameter that were embedded in 1% (*w*/*v*) agarose gel were used. PSFs were calculated by analyzing the *z*-stack images of the beads using PSFj software ver.3 (http://www.knoplab.de/psfj/, accessed on 7 November 2018) [28].

### 4.3. Fish Husbandry

Medaka fish were maintained in freshwater tanks with a water circulating system (LABREED, IWAKI, Tokyo, Japan) at 26–28 °C under the conditions of a 14-h-light and 10-h-dark cycle (08:30–22:30 light). Fish were fed artemia larvae and a powdered diet daily. The spawned eggs were collected in the morning, and eggs were incubated at 28 °C in a dish filled with diluted artificial seawater (0.03 (*w*/*v*)%) containing methylene blue. Dechorionation of eggs was performed using a hatching enzyme before microscopy observations. The transgenic strain FLT4-EGFP, which expresses EGFP [19] in lymphatic endothelial cells, was kindly provided by Dr. Deguchi (National Institute of Advanced Industrial Science and Technology). All experiments were conducted in accordance with the Guidelines of the Safety Committee for Gene Recombination Experiments at Ehime University.

### 4.4. Preparation of Medaka Embryos

Before the observation, embryos were dechorionated using the hatching enzyme, which was provided by National BioResource Project (NBRP) Medaka. For observations, we embedded embryo samples into a 1% (*w*/*v*) agarose gel set in the sample holder, which was put into the sample chamber filled with the embryonic culture medium [29]. The number of samples used in the experiments was FLT4-EGFP × 3 for live imaging of the lymphatic vessels.

### 4.5. Image Processing

Maximum intensity projection and line selection were performed using Fiji software ver.2.0.0-rc-3 and later [30].

### 4.6. Statistical Analyses

Data are presented as box plots, with the lines of the box indicating the first quantile, median, and third quantile. Upper and lower whiskers indicate the minimum and maximum, respectively. The cross mark represents the average. The asterisk indicates a *p*-value < 0.05 with the sign test. Statistical significance was determined using the Excel software (Office 2019, Microsoft).

## Figures and Tables

**Figure 1 ijms-24-10186-f001:**
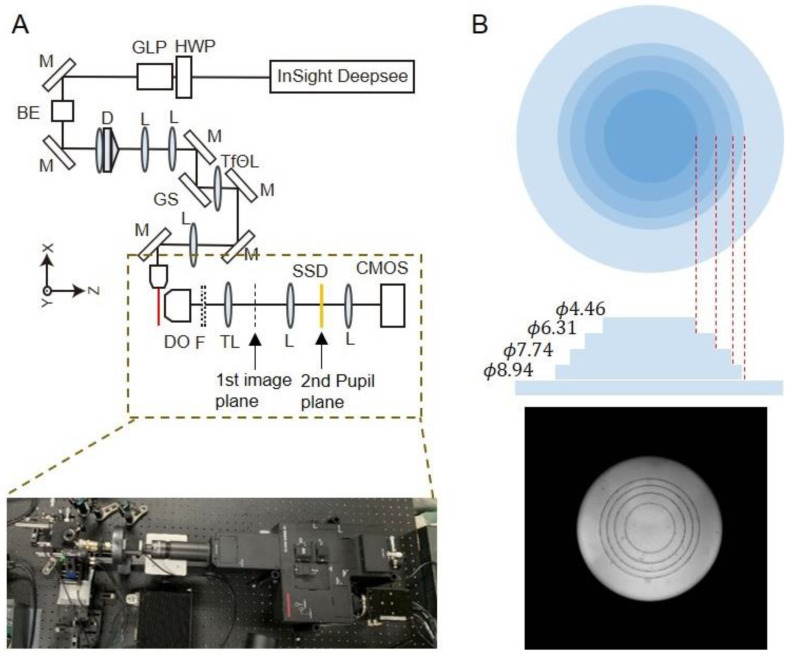
Microscope system. (**A**) Optical scheme of the light-sheet microscope system. For illumination, the Bessel beam–forming unit was introduced. A Ti—sapphire laser oscillator, Insight Deepsee (700–1300 nm wavelength)—was used as the laser source. The illumination and the detection pathways were set perpendicular to each other. A Galvano mirror scanner was introduced to generate the light-sheet, and the created optical slice was imaged by a sCMOS camera. The sample mounting holder was controlled by the motorized *xyzθ* stages. For imaging, the image-splitting optics system W-View Gemini-2C (A12801-10, Hamamatsu Photonics, Hamamatsu, Japan) was used. The extended DOF device was inserted in the second pupil plane of the optical system. Abbreviations: HWP, half wavelength plate; GLP, glan-laser polarizers; M, mirror; BE, beam expander; D, doublet; L, lens; GS, Galvano mirror scanner; TfΘL, telecentric fΘ lens; DO, detection objective lens; TL, tube lens; F, emission filter; SSD, stair-step device. (**B**) Specification of the extended DOF device. The image (bottom) was acquired at the pupil plane with a Beltran lens.

**Figure 2 ijms-24-10186-f002:**
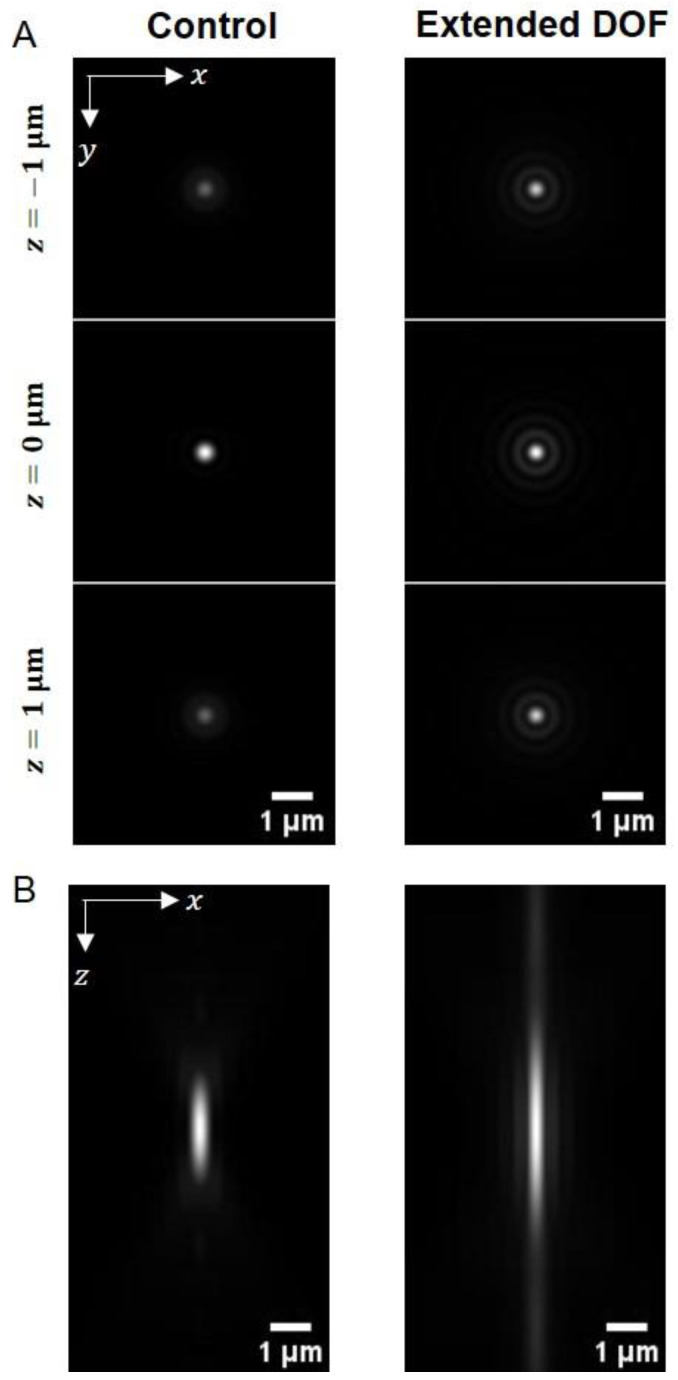
Computer-simulated PSFs for the microscope system with and without the extended DOF device. (**A**) The *xy*-plane images of the control (without the device) and the extended DOF (with the device) PSFs for different *z*-position (*z* = −1, 0, 1 μm). (**B**) Maximum intensity projection images of the control and the extended DOF PSFs onto the *xz*-plane.

**Figure 3 ijms-24-10186-f003:**
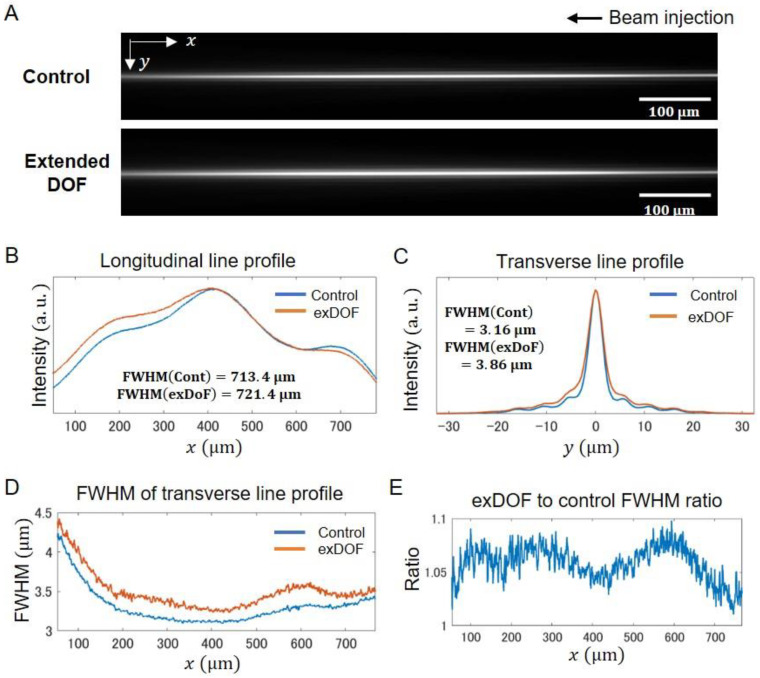
Comparisons of the beam line profiles between the control and the extended DOF system. (**A**) Laser line images of the control and the extended DOF system. Fluorescent signals emitted by the fluorescein solution (10 μg/mL) were imaged through the optical system. (**B**) Quantified line intensity profiles along the longitudinal (*x*) direction. The numerical values for signal extents (FWHMs) are shown. (**C**) Quantified line intensity profiles along the transverse (*y*) direction at the beam focus. The numerical values for signal extents (FWHMs) are shown. (**D**) Estimated FWHMs of the transverse line profiles as a function of the longitudinal axis. (**E**) The ratio of the extended DOF to control FWHMs (transverse) as a function of the longitudinal axis.

**Figure 4 ijms-24-10186-f004:**
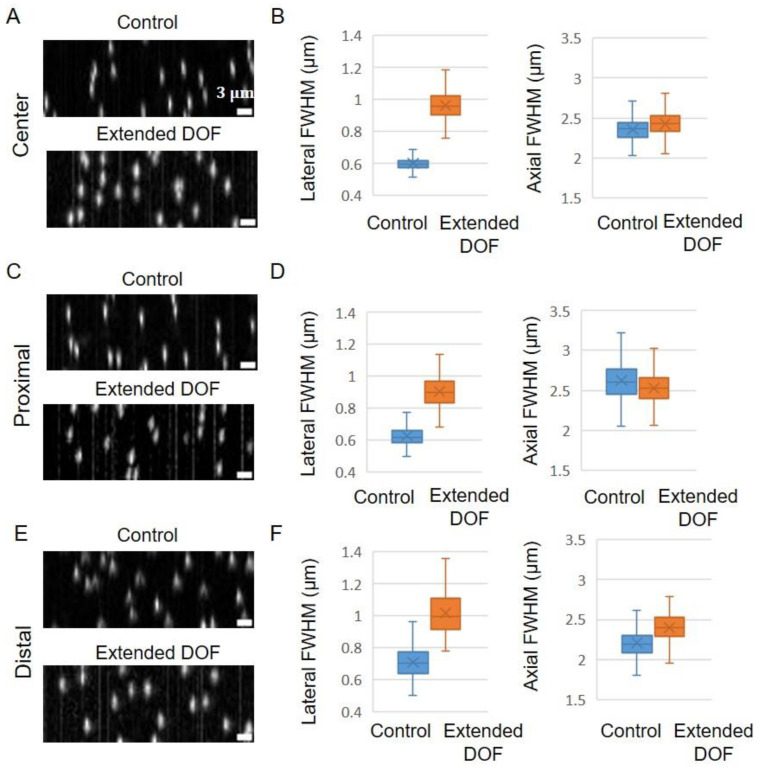
Measurement of the beads PSFs. The cropped images were created in the center, proximal, and distal areas in the images. (**A**,**C**,**E**) Fluorescent bead images were measured by the system with and without the device. Maximum intensity projection along the *y* direction is performed for visualization on the *xz* plane. (**B**,**D**,**F**) The calculated FWHM values for the lateral and axial directions. The PSF calculation employed over a hundred independent beads as data points. In the box plots, the box lines indicate the first quantile, median, and third quantile. Lower and upper whiskers indicate the minimum and maximum, respectively. The cross mark represents the average.

**Figure 5 ijms-24-10186-f005:**
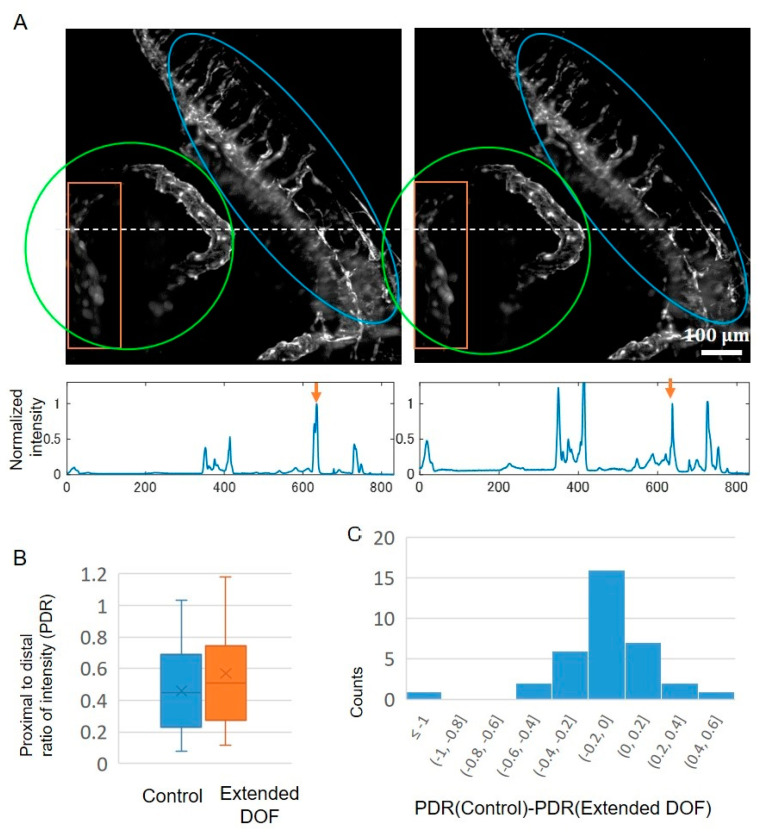
In vivo imaging application. (**A**) Comparison of the whole embryo images with and without the device. Embryos of the FLT4-EGFP strain, which expresses EGFP in lymphatic vessels, were observed. Images shown were acquired using the same embryo. Orange rectangles at the distal site of the illumination indicate the area where significant differences appear in the comparison of the images. The blue circle indicates the area of the lymphatic vessel in the body trunk, while the green circle indicates the area of the veins surrounding the yolk. Graphs beneath the images were line intensity profiles along the white dashed lines. The images were subjected to maximum intensity projections. (**B**) Proximal-to-distal intensity ratio quantified by the images. For the analysis, a total of 35 independent line profiles were extracted from *n* = 3 biologically independent embryos. In the box plots, the lines of the box indicate the first quantile, median, and third quantile. The upper and lower whiskers indicate the minimum and maximum, respectively. The cross mark represents the average. The asterisk indicates a *p*-value < 0.05 with a one-sided *t*-test. (**C**) Histogram of differences in the proximal-to-distal intensity ratio between the control and the extended DOF (control−exDOF). Sign test *p* < 0.05. Exposure time was 100 ms, *z*-interval was 2 μm, and *xy* 0.8 μm/pixel (1024 × 1024 pixels).

## Data Availability

The datasets generated during and/or analyzed during the current study are available from the corresponding author on reasonable request.

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
