# Peer review of "Extended Depth of Focus Two-Photon Light-Sheet Microscopy for In Vivo Fluorescence Imaging of Large Multicellular Organisms at Cellular Resolution"

_ijms, 2023, doi:10.3390/ijms241210186_

Round 1
Reviewer 1 Report
In this manuscript, the authors applied a stair-step device to extend the depth of focus (DOF) of a two-photon Bessel beam light-sheet microscope to cover the light-sheet thickness and achieve high-resolution live imaging. The manuscript is well-written and explains the technical detail clearly. I think it is suitable to publish and will attract the attention of the related field. Here are a few minor comments.
1. Figure S1A. I think z=-um was mislabeled.
2. The authors may give more detailed descriptions of the simulation of the PSFs and show the difference with and without extended DOF. Also, there is no full name of PSF shown in the manuscript.
Author Response
We would like to express our sincere appreciation for your comments. Based on the comments, we have made corrections to the text. The revised manuscript includes modifications indicated in red text.
Point-by-Point response:
1. Figure S1A. I think z=-um was mislabeled.
> This has been corrected.
2. The authors may give more detailed descriptions of the simulation of the PSFs and show the difference with and without extended DOF. Also, there is no full name of PSF shown in the manuscript.
> For detailed descriptions of the PSF simulation, we have added Appendix A: Computer simulation of the point spread function on line 434 (page 14). In this appendix, we describe how the PSF is calculated both with and without the extended DOF device. The simulated results are presented in Figure 2, as in the first version of the manuscript. Additionally, we have included the full name of the PSF in the text on line 112 (page 3).
Reviewer 2 Report
This manuscript utilizes the five-layer zone plate from Hamamatsu Photonics (A12802-35-100) to extend the depth of field of a high numerical objective lens to overcome the mismatch of the illumination and detection of their previously reported two-photon light sheet microscope. The employment of the extension device successfully increases the axial detection range. Consequently, the full-intensity profile of the illumination can be used to excite the sample, and more fluorescence signal is retrieved. The overall SNR with the device is improved with a slight reduction of the lateral resolution. Comparing the other DOF-extending methods, this study provides a concise and easy setup approach to enhance the light collection efficiency in light sheet imaging. I recommend the acceptance of this manuscript after the minor questions below are resolved.
Minor questions:
-
In the figure legend of Figure 1., the explanation of the abbreviation SSD is missing.
-
Figure 3E. is not cited in the content, and the corresponding discussion is lacking.
-
In Figure 4, It would be more consistent to use proximal, center, and distal to represent the left, right, and center parts in the image.
-
Figure 5 and the description from line 242 to line 246. The authors intend to show that in the extended DOF case, the distal side intensity has an improvement over the control. However, the extended DOF case has a higher proximal to distal ratio of intensity than the control, which might lead the readers to misinterpret the data. I believe the higher PDR of the extended case is due to the overall improved intensity, as shown in the line profile of Fig 1A. Therefore I would suggest the author consider using other metrics to exhibit the improved intensity in the distal area.
- Why this product, A12802-35-100 is used ? not A12802-35-040 nor A12802-35-065,
Author Response
We would like to express our sincere appreciation for your constructive comments. Based on your comments, we have made revisions to the text and figures. The modified sentences are highlighted as red-colored text in the manuscript.
Point-by-Point response:
Minor questions:
In the figure legend of Figure 1., the explanation of the abbreviation SSD is missing.
> SSD is an abbreviation of the stair-step device. This has been corrected.
Figure 3E. is not cited in the content, and the corresponding discussion is lacking.
> Figure 3E citation has been added on line 182 (page 6), and the corresponding discussion is included in the text following it “This is in good agreement …”.
In Figure 4, It would be more consistent to use proximal, center, and distal to represent the left, right, and center parts in the image.
> This has been corrected as suggested.
Figure 5 and the description from line 242 to line 246. The authors intend to show that in the extended DOF case, the distal side intensity has an improvement over the control. However, the extended DOF case has a higher proximal to distal ratio of intensity than the control, which might lead the readers to misinterpret the data. I believe the higher PDR of the extended case is due to the overall improved intensity, as shown in the line profile of Fig 1A. Therefore, I would suggest the author consider using other metrics to exhibit the improved intensity in the distal area.
> Thank you for your comment. This is an important point. As mentioned by the reviewer, the overall intensity improvement for the extended DOF may largely contributes to the proximal to distal intensity ratio, although morphological changes are observed in Figure 5A which suggest beam modulation in living samples. Therefore, we have revised the description to clarify that the objective of the PDR analysis is to confirm the existence of effective fluorescence correction in in vivo imaging at the distal side, similar to the fluorescence solution experiment. Please refer to the text on line 250 (page 9).
Why this product, A12802-35-100 is used ? not A12802-35-040 nor A12802-35-065,
> For detailed descriptions of the PSF simulation, we have added Appendix A: Computer simulation of the point spread function on line 434 (page 14). In this appendix, we have also provided the reason why the product A12802-35-100 is used on line 451 (page 14). This product is designed for larger pupil diameters, specifically those for greater than 10 mm, while other models are designed for shorter pupil diameter. In our case, with the pupil diameter of 13 mm, this product is most suitable for our system.
Reviewer 3 Report
The manuscript ijms-2432536 by Takashi Saitou and Takeshi Imamura presents a two-photon digitally scanned light-sheet microscope tailored for live cell imaging over approximately 1 mm field of view (FOV). The authors demonstrate that by adding a stair-step device composed of five-layer annular zones they achieve a two-fold DOF extension, which is enough to cover the light-sheet thickness. They characterize the resolution of the modified system and compare it with the original one. Finally, they apply this system to in vivo imaging of medaka fish and find an improvement of image quality over the FOV.
My overall view about the manuscript is positive, it is well written and clearly presented, only few comments are provided in the attached PDF. After this minor revision, it would become suitable for publication in IJMS.

Author Response
We would like to express our sincere appreciation for your valuable comments. Based on your comments, we have made revisions to the manuscript. Please refer to the attached file for our response. The modified sentences are highlighted as red-colored text in the manuscript.
